# Unraveling EFL Teacher Buoyancy in Online Teaching: An Ecological Perspective

**DOI:** 10.3390/ijerph20010613

**Published:** 2022-12-29

**Authors:** Honggang Liu, Siyu Duan, Wenxiu Chu

**Affiliations:** 1School of Foreign Languages, Soochow University, Suzhou 215006, China; 2School of Foreign Languages, Northeast Normal University, Changchun 130024, China

**Keywords:** EFL teacher buoyancy, Bronfenbrenner’s ecological systems theory, dynamic interaction, socio-ecological product, qualitative study

## Abstract

Due to the COVID-19 pandemic, online teaching became a significant method at different levels of education across the globe. The transition from traditional offline to online educational environments brought new challenges for language teachers. Buoyancy plays a crucial role for teachers to bounce back from challenging situations. However, there is a scarcity of empirical research on language teacher buoyancy in online contexts from an ecological perspective which is conducive to unfolding the complex and dynamic nature of buoyancy. To fill this gap, the current study utilized a qualitative research method to investigate the factors influencing English teacher buoyancy in online teaching and how they shape and exercise buoyancy in their negotiation with different ecological systems in online teaching guided by Bronfenbrenner’s ecological systems theory. The findings revealed that teachers experienced multiple challenges from different ecological systems, such as ineffective classroom interaction, work–life imbalance, heavy workload, and higher school requirements. Additionally, teacher buoyancy was shaped by the dynamic interaction between teachers and ecological systems and was not only viewed as the individual’s ability but as a socio-ecological product. Based on the above findings, the paper provides some implications for developing and researching language teacher buoyancy in the future.

## 1. Introduction

Language teaching is often viewed as a highly demanding profession [1] and faces many dangers [2]. Previous studies have identified numerous challenges that may arise in the course of language teachers’ work, such as heavy workloads, insufficient support, high turnover, and students’ misbehavior [3,4] and the accumulation of these daily exertions may become an invisible but long-term source of pressure for language teachers [5]. In facing the challenges and difficulties of daily life, language teachers should be capable of successfully coping with the setbacks typical of the daily teaching practice, which is termed teacher buoyancy [6,7,8]. In the special education context caused by the COVID-19 pandemic, online teaching came out as the major teaching mode [9,10], which posed many new requirements and challenges for English as a foreign language (EFL) teachers, such as the demand for improving teaching competence and quality, and insufficient classroom interaction in online teaching [5,11]. The exploration of teacher buoyancy in general education gives us an insight into how teachers negotiate and overcome the setbacks and challenges that are part of their daily working life, which is a pivotal determinant of teachers’ professional development and teaching quality. To the best of our knowledge, studies on teacher buoyancy in general education explored its inner structure [6,8,12], relationships with other demographic and psychological variables [12,13,14], and the role of teacher buoyancy in their teaching effectiveness and well-being [13,15]. However, research happening in the online teaching environment is mainly restricted to the supportive effects of teachers’ creative resources on adaptability and buoyancy in the presence of the COVID-19 pandemic [16], and very few scholars give much consideration to the change and development of language teacher buoyancy during their interaction with the ecological systems in which they are embedded, especially during this difficult period. Research on teacher buoyancy to date in the online teaching setting is still at an under-explored stage [16].

With the popularity of the ecological perspective in the field of language teacher education and development, the ecological systems theory proposed by Bronfenbrenner [17,18] not only focuses on the role of different layers of the ecological environment in shaping and influencing the development of human beings, but highlights the value of the developing persons, and their dynamic interconnection, interdependence, and interaction with the environment. It is beneficial to demystify the complex teaching world and how teachers interact and negotiate with the context where they live to promote their psychological and professional development [4,5]. Actually, as agentic and developing persons living and working in different social systems, language teachers are faced with a range of effects of ecological factors on their psychological and professional growth [4,5], and their buoyancy contributes to motivating their ability to respond to these impacts and overcome setbacks and adversities in the daily professional world. Therefore, it is necessary to investigate language teachers’ buoyancy in their interaction with surrounding significant persons and events in the online educational context from an ecological perspective. The exploration into this topic will not only help delve into their perceived challenges and their buoyant performance in their daily online teaching practice, but also offer inspiration for promoting the general research status of teacher buoyancy and enrich the much-needed research on language teacher psychology [19,20]. Simultaneously, research on language teacher buoyancy from an ecological perspective will also shed valuable insights into the research on foreign language online teaching psychology in the post-pandemic era and provide critical implications for the establishment of a harmonious ecological environment for developing teacher buoyancy, and thus for the improvement of the online teaching quality. Against this backdrop, this study conducts a qualitative exploration of the factors influencing EFL teacher buoyancy in online teaching and how EFL teachers shape and exercise buoyancy in their negotiation with different ecological systems in online teaching from an ecological perspective.

## 2. Literature Review

### 2.1. Defining EFL Teacher Buoyancy

Some scholars [12] are the proponents of the concept of workplace buoyancy which reflects the employee’s ability to manage setbacks, challenges, adversity, and pressure effectively. They conceptualized workplace buoyancy as, “employees’ ability to effectively deal with setbacks, challenges, adversities, and pressures in the workplace setting” [12] (p. 172). There are some similarities between buoyancy and resilience [7,21,22], both of which focus on an individual’s capacity to cope with difficulties, but resilience addresses the personal capacity to deal with acute and chronic setbacks and pressures in highly adverse situations, whereas buoyancy highlights the features of “everyday resilience”, and more centers on the ability of the individual in specific situations (e.g., school) to successfully cope with the ups and downs of daily life [23]. Zhang [7] further identified characteristics of buoyancy, such as proactive responses to challenges and a focus on ordinary people rather than people in highly adverse situations.

Teacher buoyancy is considered as the teacher’s ability to respond positively to daily challenges, frustrations, and difficulties, and to access potential resources and advantages to cope with recurring problems in the school environment [6,8]. Additionally, the notion of teacher buoyancy is actually different from teacher agency which more emphasizes the teacher’s active efforts or intentional authority to make choices and act accordingly in the context where he or she operates [24,25,26]. However, teacher buoyancy focuses on the individual’s capacity to successfully cope with adversities that are typical of the ordinary course of professional life [6,15]. Inspired by previous studies, we also conceptualize teacher buoyancy as the teacher’s ability to respond to and manage the setbacks and challenges routinely encountered in daily teaching practices. Teachers who are constantly exposed to challenges need to be equipped with such a regulatory mechanism so as to maintain healthy mental functioning and to be capable of working with ease. The online teaching setting presents difficulties that may not appear in the traditional face-to-face teaching environment, raising new challenges for teachers, such as the lack of knowledge of online teaching design, the complexity of network operations, and the anxiety caused by the instability of network signals and home quarantine during the pandemic [5]. In this special online context, teacher buoyancy can serve as a psychological regulation mechanism that teachers demonstrate to cope with daily difficulties and setbacks in online teaching settings, and the ability to maintain a positive mindset.

### 2.2. Unpacking the Profile of Teacher Buoyancy

Aside from the definition of teacher buoyancy mentioned above, researchers in the field of educational psychology have conducted some exploration on this topic. scholars have attempted to examine the structure [6,8] and influencing factors [12,13] of teacher buoyancy as well as its relationships with psychological variables, such as teachers’ work-related well-being and engagement [15], well-being and creativity [16], and self-efficacy and burnout [14]. For example, Wong et al. [6] developed a teacher buoyancy scale and extracted a five-dimensional structure, namely, coping with difficulties, bouncing back cognitively and emotionally, working hard and appraising difficulties positively, caring for one’s well-being, and striving for professional growth. The study also found that primary and secondary school teachers did not perceive themselves as capable of bouncing back cognitively and emotionally in schoolwork, and they were less likely to believe they could take care of their overall well-being compared with using strategies to address challenges in work. Moreover, Tang et al. [8] used a mixed-methods study to explore the structure of early career teachers’ buoyancy and how they employed personal and contextual resources to deal with everyday challenges in their professional world. The results revealed that teacher buoyancy was composed of six dimensions: bouncing back emotionally and taking care of one’s well-being, cognitive reframing, coping with difficulties, striving for professional growth, support from family and friends, and support from the work context. The former four dimensions were categorized as personal resources, and the other two belonged to contextual resources. It was also found that teachers drew upon their emotional, cognitive, and behavioral emphases of personal resources, and developed social–professional relationships within and outside the school context for affective and cognitive support, thus helping make immediate to short-term responses to everyday teaching challenges.

In addition, the relationships between teacher buoyancy and other factors were also given some attention. For example, Parker and Martin [15] explored the predictive effect of teacher buoyancy on teachers’ work-related well-being and engagement. The study revealed that teacher buoyancy was a significant predictor of teacher well-being and engagement, with the increase in teacher buoyancy contributing to teachers’ enjoyment of work, participation, and positive career aspirations. Moreover, this study highlighted the need to consider teacher-level interventions in the enhancement of teacher well-being and working engagement which could not be effectively achieved by the organizational-level and school-level efforts alone.

Collie et al. [13] examined how two personal resources among secondary school teachers (adaptability and buoyancy) are predicted by two contextual factors (perceived autonomy-support, time pressure), and their links with three work-related outcomes (organizational commitment, extra-role behavior, and failure avoidance motivation) guided by job demands-resources theory. Their findings revealed that perceived autonomy support and time pressure were significantly associated with adaptability and buoyancy. Moreover, buoyancy was associated with lower failure avoidance and served as a mediator of the association between time pressure and failure avoidance. Ding and He [14] explored the relationship between Chinese EFL teachers’ buoyancy, self-efficacy, and burnout using a quantitative research design, and found a positive correlation between teachers’ academic buoyancy and self-efficacy, but a negative association between their buoyancy and burnout. Martin and Marsh [12] identified gender and age differences in teacher buoyancy and illustrated that male teachers were significantly more buoyant than female teachers, and elderly teachers were more buoyant than younger teachers. They revealed that the enhancement of teachers’ work engagement, job enjoyment, positive intentions, and work engagement outside the workplace contributed to teacher buoyancy.

Additionally, scant attention has been paid to the research on teacher buoyancy in online teaching contexts. Among the few studies, Anderson et al. [16] examined how teachers’ creative resources (creative beliefs and affective and environmental support for creativity) support their adaptability and buoyancy in the face of the COVID-19 school shutdown. They found that COVID-19 and distance learning brought new challenges and greater stress to teachers and dampened their capacity to build a supportive connection with students. Teacher creative self-efficacy might help teachers remain buoyant in the presence of challenges. A creative growth mindset might facilitate a positive effect for teachers, and a reduction in creative anxiety might make it easier to avoid the negative effect and traumatic stress. Moreover, environmental support and encouragement for creativity in schools might be key to motivating teacher adaptability and buoyancy during the pandemic.

By and large, research on teacher buoyancy appears to cover extensive themes, with a focus on the internal structure and influencing factors, as well as its correlations with other psychological variables. However, there has been a scarcity of empirical and theoretical studies on language teacher buoyancy. Most of the existing research has been conducted in traditional face-to-face teaching settings, and few studies shift their focus to investigate the development of the buoyancy of EFL teachers in the online teaching context. However, EFL teachers are experiencing more diverse and intense teaching difficulties and pressure due to the COVID-19 pandemic, such as health concerns and insufficient technological support, inadequate technological pedagogical and content knowledge, and insufficient teacher–student interaction [5]. Exploring EFL teacher buoyancy is crucial to reveal the pressures and challenges teachers encounter in online teaching practice during the pandemic and unveiling their ways to cope with daily adversities and risks. Notable, teachers, as developing persons, can interact with the ecology and use available resources to address external stressful events, thus promoting their professional development [4]. Bronfenbrenner’s ecological systems theory [17,18], as a highly recognized and widely utilized ecological theory, provides insightful theoretical guidance for the exploration of the dynamics between teachers and their ecological environments and the development of EFL teacher buoyancy in the online teaching context.

## 3. Bronfenbrenner’s Ecological Systems Theory

With the ecological turn in language teacher education, as one of the important ecological theories, Bronfenbrenner’s ecological systems theory/model has received growing recognition and has been widely utilized in research on language teacher professional quality [4], anxiety [5], resilience [27], emotion [28], and personal style [29]. Bronfenbrenner’s ecological systems theory considers the dynamic interaction between people and the environment, viewing the individual as a developing person with unique force, resource, and demand features, and unravels the different ecological contexts where people operate [17,18]. Therefore, this theory offers significant insight into the investigation of the formation and development of EFL teacher buoyancy in their interaction with the ecological environments. This model is a set of nested ecological systems and involves the microsystem, mesosystem, exosystem, and macrosystem.

Specifically, the microsystem is regarded as a pattern of activities and roles experienced by the developing person in the immediate setting with specific physical, social, and symbolic features [18]. Inspired by this definition, the ecological factors which are closely related to teachers’ online teaching experiences and activities in the immediate online setting are situated in the microsystem, such as teacher’s psychological feelings, teaching experiences related to the online course, the ability to operate the network equipment, and teacher–student interaction in the online classroom. The mesosystem in Bronfenbrenner’s ecological systems theory is often viewed as a relational system [30] focusing on the interrelations among two or more settings in which the developing person becomes an active participant [18]. Previous studies have also categorized interpersonal relationships (i.e., teacher–leader relationships and parent–teacher relationships) into the mesosystem. Therefore, this study classified various relationships beyond the immediate online teaching setting into the mesosystem, such as the school–teacher relationship, parent–teacher relationship, and student–teacher relationship. The exosystem involves linkages and processes taking place between two or more settings, at least one of which does not contain the developing person, but the events in the settings indirectly affect the activities and experiences of the person happening in the immediate environment [18]. For example, local policies and teaching regulations about online teaching are usually not made by ordinary teachers but do exert a significant impact on their online teaching practice. The macrosystem is viewed as a societal blueprint in a given culture, subculture, and other broader social contexts [18], such as beliefs, resources, and lifestyle. In the current study, the macrosystem primarily involves the situation of the COVID-19 pandemic and educational policies such as “classes suspended but learning continues”. Inspired by previous studies on teacher buoyancy and Bronfenbrenner’s ecological systems theory, this study attempts to explore challenges that EFL teachers perceive and how they shape and exert their buoyancy in the negotiation with different ecological systems in the Chinese online teaching context.

## 4. Methodology

### 4.1. Research Context and Participants

Due to the spread of the COVID-19 pandemic, the Ministry of Education in China issued the educational policy to ensure the stable and smooth development of teaching schedules through online platforms and resources. During this special time, online teaching became an extremely important approach to teaching in China. However, the sudden transition from traditional offline to online educational environments brought new challenges for teaching staff. Most of them were not trained to teach online before and lacked the relevant knowledge and skills in the technology-based educational environment. Although teachers were trained on how to use the network teaching platform during this period, there were still an array of teaching adversities, causing significant uncertainty and pressure on their psychological and professional development. In this context, this study used purposeful sampling to select volunteers to participate in the following semi-structured interview in order to unpack the challenges teachers encountered in online teaching and the development of their buoyancy to manage these adversities. In this process, the authors considered participants’ differences in gender, educational background, and years of teaching experience. Nine volunteers who showed great willingness to share their feelings and experiences about online teaching were invited to participate in this study through social media platforms (i.e., Ding Talk and Tencent Meeting). Among them, five were female teachers and four were male teachers, five had achieved a bachelor’s degrees and four had achieved a master’s degrees, and their years of teaching experience ranged from 4 to 30 years. More details can be seen in Table 1.

### 4.2. Data Collection

Semi-structured interviews were utilized to capture challenges teachers encountered in online teaching practice and how they shape and exert their buoyancy to respond to daily challenges. The semi-structured interview offers, “the opportunity to step into the mind of another person, to see and experience the world as they do themselves” [31] (p. 9). Therefore, we used this kind of interview to capture challenges teachers encountered in online teaching practice and how they shape and exert their buoyancy to respond to daily challenges. The interviews followed an interview protocol to avoid straying from the research topic [32]. The following main interview questions were involved in this study. For example, “Compared with traditional face-to-face teaching, are there any changes or challenges during your online teaching? What impact do these changes have on your professional lives? How do you deal with these challenges? Are there any resources available in the ecological environments where you live that help you address these challenges?”. Participants were informed of the research purposes before they voluntarily participated in this study, and they were also assured that the data would be kept confidential and only used for academic research. Each interview was conducted in Mandarin Chinese and lasted for about 30 min and was audio-recorded after the permission of the participants.

### 4.3. Data Analysis

In the study, qualitative content analysis [33] was employed to analyze the qualitative data stored in NVivo 12. Specifically, an inductive approach was employed to construct the theme from the data [34]. Recordings of the interviews were first transcribed and checked by the participants to ensure transcription accuracy, and then carefully reviewed by authors to identify codes related to ecological factors that shape and activate teachers’ buoyancy to navigate challenges with the guidance of Bronfenbrenner’s ecological systems theory. For example, the data depicting the factors that deeply influenced teachers’ interconnection and interaction with ecological elements in the immediate settings and activated the fluctuation and development of teacher buoyancy, were categorized into the microsystem, such as teachers’ online teaching practice, the technological pedagogical and content knowledge (TPACK), which describes the relations and interaction between teachers’ knowledge of technology, pedagogy and the subject matter [35]. Likewise, the relationships between EFL teachers and surrounding significant persons (i.e., school leaders, colleagues, students’ parents, and family) that influence how they respond to the teaching setbacks taking place in the immediate setting were encapsulated in the mesosystem, which was usually regarded as the relational system [18,30]. Additionally, educational demands and policies which were proposed by local educational authorities and school leaders rather than EFL teachers themselves, but deeply affected their teaching practice, could be categorized into the exosystem. Finally, elements such as the educational policy of “classes suspended but learning continues” launched by the Ministry of Education in China and social emergent events (i.e., the COVID-19 pandemic) belonged to the macrosystem. The authors independently coded the interview transcripts, and then discussed the different codes with the guidance of the analytical framework. For example, the excerpt “I usually communicated with my colleagues and found out what stressful events they encountered, and they were willing to tell me about their experiences (T1)” highlighted the role of colleagues in helping the teacher respond to online teaching setbacks. They were significant persons closely associated with the teacher himself or herself. Therefore, the relationship between colleagues and the teacher was coded in the mesosystem. In addition to this coding process, participants were also invited to check our findings to guarantee the reliability of the study.

## 5. Findings and Discussion

Based on the analysis of the interview data guided by Bronfenbrenner’s ecological systems theory, this study unpacked the formation and development of EFL teacher buoyancy from different ecological systems and constructed the nested ecological model for EFL teacher buoyancy in online teaching settings.

### 5.1. Microsystem: EFL Teacher Buoyancy Triggered by the Immediate Online Teaching Context

The microsystem is the innermost layer of the ecological systems and focuses on a pattern of activities and roles experienced by the developing person in the immediate settings with specific physical, social, and symbolic features [18]. In the online teaching environment, the microsystem primarily involved the teachers’ TPACK and teaching competence, online lesson, classroom interaction, and other factors that took place in the immediate online class.

Most participants reported a lack of immediacy in distance learning which narrowed the avenues for teachers to instantly check the effectiveness of their online teaching. In the traditional face-to-face teaching context, teachers could observe the students’ learning performance and adjust their teaching style and pace in light of students’ reactions at any time. However, in distance teaching, it was particularly difficult for teachers to perceive students’ responses through the screen, and the immediacy of interaction with students could not be guaranteed. Distance teaching brought new challenges to teachers and dampened their capacity to build a supportive connection with students [5,16]. As T6 and T1 revealed, online teaching, “was more like a dialogue between the teacher and individual students rather than a whole group of students” (T6), and “it was not convenient to interact with students in online classes, and sometimes students were not willing to answer questions, so online teaching was not as effective as offline teaching” (T1). Students might give uninformative answers in online classes. As T4 reported, “some students would just follow other students and give the same answers or emojis, such as ‘yes’, ‘I got it’, ‘😊’” (T4), and it was hard for the teacher to judge whether students really understood the teaching content.

Faced with these challenges in online teaching, teachers were not powerless, instead, they could employ their life capital [36] and exert proactive agency to adopt certain coping strategies to respond to these difficulties. This was evident in the interview excerpts below.


*At the beginning of online teaching, I tried to get my online classes in place, although I had only learned a little about the use of online platforms under the school teaching training […] I believed teachers should fulfill their educational duties and educate students well and pay attention to their mental health during this difficult period. However, it is hard to achieve such an educational goal and there are so many factors influencing my teaching, so I stopped giving too much pressure on myself and tried my best to improve my online teaching competence.*
(T2)

T2 was not experienced in teaching online and had mastered relevant knowledge from the school training. In the microsystem, the lack of TPACK and online teaching experience resulted in severe stress, which prompted her to explore approaches to address the challenges. She was also an idealist in her educational beliefs and values and started her online teaching with a vision of cultivating all students well and caring about their physical and mental health. However, confronted with various obstacles, she recognized the gap between the teacher’s ideal self [37] and teaching reality. Nevertheless, she managed to actively adjust her mindset to reduce her stress and enhance her competence in online teaching, gradually reconciling with herself and smoothly responding to these perceived setbacks. Previous studies also highlighted buoyant teachers could accept difficulties and bounce back emotionally from daily challenges, perceive difficulties as opportunities that encourage teachers to improve their teaching quality and maintain their hope and enthusiasm for teaching [8,14].

When faced with the challenges emanating from online teaching, T3 also felt exhausted due to the lack of online teaching experience, ineffective classroom interaction, and the spread of the pandemic. Those challenges were also found in the previous research on online teaching during the pandemic [5,38,39]. However, she took action to activate her buoyancy for managing the pressure caused by online teaching.


*Online teaching was quite demanding, especially when you had no teaching experience. Before we started the online classes, the school gave us training on how to use the online platform and I just learned some basic operations. During online classes, I was worried that network signal problems would cause a lag in students’ responses. I could not do anything but feel anxious. This greatly diminished teaching and learning quality […] Therefore, I read more books and browsed on WeChat moments and posts on some apps. However, I found it necessary to sift through this information because some of the posts were inspirational articles that gave your verbal encouragement. It was still better to master more practical knowledge about the design of online teaching activities and strategies to cope with problems in English classes.*
(T3)

Both T3 and T2 had no experience in online teaching. However, although difficulties in teaching existed, such as internet lag and low teaching efficiency, T3 sought to develop her expertise through browsing online posts and autonomous learning and take practical actions to free herself from the constraints of the problems. She was also reflective, recognizing the role of the inspirational article in briefly inspiring individual teachers in the face of teaching challenges, but also realizing that a simple psychological incentive was not conducive to long-term professional development. She argued that teachers need to adapt their own teaching designs to suit the requirements of online teaching and improve their professional competence. This indicated that her buoyancy was not limited to the positive psychological mood, but rather reflected in the behavioral aspect of harnessing personal and contextual resources to respond to and cope with setbacks [6,8] in the presence of everyday online teaching pressures.

As mentioned above, EFL teachers’ lack of experience in online teaching practice and training coupled with the problems in the online teaching environment (i.e., poor teaching effectiveness, unstable network signal), placed them in a predicament of online teaching. However, buoyant teachers were able to perform psychological and behavioral self-regulation in response to their perceived adversities and setbacks [8], thus shaping and developing their stronger buoyancy for maintaining a positive mindset, negotiating, and addressing the ups and downs with ease.

### 5.2. Mesosystem: EFL Teacher Buoyancy Intertwined with the Role Conflict and Interaction with Significant Others

The mesosystem is a relational system, highlighting the interrelations among two or more settings in which the developing person becomes an active participant [18]. In this study, the relationships beyond the immediate classroom context were emphasized and reflected in the mesosystem of this study, such as the relationships between teachers and their students, colleagues, school leaders, and students’ parents. Notably, these significant others provided affordance to facilitate teachers’ teaching but also imposed new demands on teachers [4], especially during the period of special online teaching. For example, school leaders actively offered teachers educational resources but endowed them with new pressure brought by more complicated requirements about online teaching and classroom management, and supervision of students’ psychological and physical health (i.e., T1, T2 and T3); Parents gave verbal support and encouragement to teachers’ work, but also added considerable extra pressure to teachers due to frequent contact with teachers concerning students’ academic problems (i.e., T4 and T9). However, these EFL teachers, as developing persons with life capital [36], did not avoid the difficulties in this particular context, instead, they endeavored to respond to the challenges by seeking a variety of supportive resources in the ecological systems, and gradually recovered and thrived from the adversities.

The exchange of thoughts, emotions and teaching experience among teachers was an indispensable activity in their professional development, which served as an effective approach for teachers to alleviate pressure, particularly in the face of teaching and life challenges. For example, T1 and T2 mentioned how they negotiated with the surrounding people to regulate their negative emotions and achieve mental recovery and development.


*I usually communicated with my colleagues and found out what stressful events they encountered, and they were willing to tell me about their experiences […] My problem-solving ability was also improving in this process, so I thought the most important point was to look forward and seek solutions instead of sulking or being frustrated without taking any action.*
(T1)


*It was much more difficult for English teachers to interact with their students in the online educational context. The English teaching plan was more likely to be delayed if we spent much time interacting with them. I talked to other teachers to learn how they taught at some time, and then felt better and tried my best to address those problems.*
(T2)

As these excerpts revealed, when faced with the dilemma of inattentive students or lack of teacher–student interaction, T1 promoted his psychological and spiritual recovery through communication with colleagues. This mutual support in the social network among colleagues alleviated T1′s anxiety. At the same time, he reflected on his own shortcomings and learned more solutions from his colleagues’ experience to respond to stressful events. T2 also exchanged teaching experience with her colleagues which seemed to be a feasible way to regulate emotion and she proactively thought about coping strategies. Both T1 and T2 resorted to the feedback and support from their colleagues in the process of addressing teaching challenges they encountered and gradually worked their way out of the online teaching dilemmas, thus developing buoyancy to cope with setbacks [6,8] caused by students’ low levels of motivation and engagement and ineffective teacher–student interaction. Different from T1, T5′s teaching buoyancy primarily stemmed from the approbation of her work from students and their parents.


*When I received positive feedback from my students, or a long message from their parents to express their gratitude, I felt that my efforts were not in vain. I found my dedication was rewarded and realized the meaning and value of being a teacher. It gave me the motivation to continue to experience tiredness but happiness in the demanding work.*
(T5)

T5 received spiritual relief from the recognition from students and their parents and further experienced a sense of purpose and meaning as a teacher. It increased her teaching buoyancy to overcome the demanding challenges of online teaching. The excerpt from T5 also demonstrated the important influence of significant others on teacher development [4,5] and suggested that the sources of teacher buoyancy in the school teaching context might involve multiple factors that were closely associated with the teacher, such as support from significant others such as colleagues, students, parents, and their own positive psychological suggestions.

The online teaching caused by the pandemic imposed a lot of redundant work on teachers [5,11], such as supervising the daily health registration, checking students’ homework on the Internet, forwarding various notices in WeChat groups, and answering parents’ queries. These tasks undoubtedly increased teachers’ work pressure.


*Due to the outbreak of the pandemic, I did not just prepare and teach online lessons, but also had to do things beyond our teaching tasks, like class management and communication with parents. With a mobile phone in hand almost 24 h a day, I needed to reply to messages timely from parents, leaders, and students.*
(T9)


*In the past, students were at school all day and parents were actually less involved in the school activities. However, now students and parents were under the same roof every day and there was a blurred boundary among their studies, work and life during this special period. It might result in conflicts between students and their parents due to communication and academic problems, and sometimes parents came to me when they could not handle their children.*
(T4)

As illustrated in the above interview extracts, during online teaching, T9 and T4 were required to devote more time and energy to tasks assigned by the school administration and respond to requirements or questions from leaders and parents. This easily made the teachers suffer from a work–life imbalance and emotional exhaustion. However, faced with these similar challenges, T8′s responses seemed to mirror the thoughts of some teachers.


*It was important to keep in a positive mood during this particular time. My family was actually quite considerate and gave me a lot of support, and I also did something for them, such as cooking and giving my child guidance in learning […] Additionally, students’ parents gave me some verbal support at such a special time, thus giving me more confidence to face difficulties related to students’ learning.*
(T8)

During the pandemic, T8 maintained a good family relationship and gained support from his family and students’ parents about his work, thus increasing his confidence to overcome the challenges. Previous research also showed that teachers often faced challenges arising from the imbalance between life and work [4,40], but these stressful events simultaneously motivated teachers to regulate themselves, clarify their roles and take action to manage challenges, which in turn cultivated their buoyancy [6].

In summary, the interaction (i.e., support from colleagues and approbation from students and parents) and collision (i.e., family–work imbalance and conflicts between complex tasks from school and teachers’ physical tiredness) between these teachers and ecological elements in the mesosystem provided sources to shape and affect their buoyancy. Buoyant EFL teachers were able to respond to problems in the daily interaction and communication with their families, students, students’ parents as well as school leaders during the special period.

### 5.3. Exosystem: EFL Teacher Buoyancy Pertaining to the Requirement of Online Teaching

The exosystem highlights the linkages and processes taking place between two or more settings, at least one of which does not include the developing person, but the events in the setting affecting the development of the immediate setting where the person lives [18]. In this study, elements in the exosystem primarily involved the educational policies and regulations set by school authorities to facilitate the online teaching process under the influence of the national policy of “classes suspended but learning continues” in the macrosystem. These elements brought teachers new challenges and pushed them to quickly adapt to the new teaching context and respond to those changes in educational policy.

Although online teaching during the pandemic compensated for the absence of face-to-face offline teaching, it also generated heavy workloads [10,41]. Teachers had to invest much time into lesson planning, teaching, and after-school tutoring on the computer and kept in touch with students, their parents, and colleagues at any time. It easily blurred the line between professional work and personal life, and thus brought about their physical tiredness and exhaustion, and long-term tension. As T3 described:


*It was as if I had to be online 24 h a day, replying to messages from school leaders or students. The school might have new arrangements and announcements, and students might send me messages at any time […] My approach was to solve problems as effectively as possible and spend free time with my family, continuing to finish my teaching tasks. In addition, I stood or paced up and down at home as a way to exercise.*
(T3)

The challenges T3 perceived in online teaching practice were also associated with the requirements of online teaching and management proposed by school authorities in the exosystem and the influence of COVID-19 in the macrosystem. Due to the spread of the pandemic, similar to other teachers, T3 was stuck in a limited space and could not move out of her house, and was always occupied with preparing lessons and checking students’ homework online. Nevertheless, she was not overwhelmed by this context rife with a great deal of uncertainty and unsafety. Instead, she employed strategies to mitigate the negative effect caused by long working hours, motivating herself to bounce back from the difficult period and cope with daily professional tasks. Previous studies also indicated that teachers needed mental and physical recovery in the presence of setbacks [3,6] and possessed rational perceptions and solutions to current difficulties, which played an important role in promoting teachers’ psychological growth and professional development [7,42].

In online teaching, teachers gained development in their teaching mindset and practice while experiencing teaching pressure. This might be observed in the following extracts:


*The school and education departments provided us with many online teaching resources, and I could look up some teaching materials that I could use in my online classes. The more difficult the situation was, the more composed and positive I had to be. Additionally, when I encountered problems, the school leaders and colleagues were willing to give suggestions and help me.*
(T6)


*I was less nervous, although a leader came to my online class, but did not gain a great sense of achievement. Sometimes it might be better to for me perceive and address some challenges so that I could thrive in a highly demanding situation. It was no use being anxious and worried and it was better to find solutions to cope with challenges and teach students well with a more peaceful mindset.*
(T7)

From the above interviews, it was found that teachers (i.e., T6 and T7) had already adapted to teaching English online during the pandemic and maintained positive emotions to respond to the current complicated teaching context. It was the salient feature of buoyant teachers and echoed the positive orientation of buoyancy [6,14]. For example, T6 utilized the online resources provided by the education department and proactively applied useful English materials to teaching activities, and she also received support from school leaders and colleagues. T7 was not afraid of the sudden supervision of school leaders in online teaching activities and endeavored to focus on the positive side of teaching risks to activate her buoyancy in the dynamic interconnection with ecological environments. Just as stated in previous research, the pandemic forced teachers to be adaptive and try new techniques in the online teaching context to some degree [11,16].

Despite higher demands for teachers during online teaching, the schools and local education departments in the exosystem provided support for teachers to smooth their teaching processes. However, it was found that teaching requirements and local teaching regulations indirectly and implicitly prolonged the duration of online teaching. The long working hours engulfed teachers’ personal lives but also drove them to proactively or reactively learn more coping strategies. The requirement and affordance for online teaching in the exosystem also contributed to activating teachers’ buoyancy to address setbacks [6,7].

In addition, within the guidance of the national educational policy of “classes suspended but learning continues”, the online platform seemed to become the only way for teaching and learning, which also exerted an impact on teacher buoyancy. However, a direct effect of ecological factors in the macrosystem on teacher buoyancy was not identified in this study, but it was found that COVID-19 and online educational policies in the macrosystem indirectly influenced the generation and development of EFL teachers’ buoyancy through their interaction with ecological factors in the microsystem, mesosystem, and exosystem. For example, the macro policy about online teaching enacted by the national education department in the macrosystem stimulated the promulgation of local educational policies and demands. However, the teaching staff who were not qualified to formulate policies needed to respond to those new teaching requirements and echo the expectation of surrounding persons during this special period of online teaching. Therefore, teachers developed buoyancy to address teaching problems in the dynamic interaction with factors in the ecological systems in spite of severe challenges.

Taken together, the emergence and development of teacher buoyancy was the outcome of the synergetic effect between multiple ecosystems and their elements. The poor teaching effectiveness and the lack of online teaching practice and knowledge in the microsystem drove teachers to actively interact with colleagues and students’ parents. The positive home–school interaction in the mesosystem allowed teachers to successfully cope with the challenges posed by the online classes in the microsystem and the requirements from the school and local education department in the exosystem. Moreover, EFL teacher buoyancy fluctuated during online teaching, as some teachers still felt stressed but displayed everyday resilience [43] and gained more coping strategies. In turn, teachers, as developing and agentic persons [18,26], possessed multiple professional and cultural identities in their interaction with surrounding persons and events [44], and their positive attitudes and coping style also helped further develop their buoyancy in the process of interaction with different ecological systems.

Based on the above analysis, we argued teacher buoyancy was rooted in complex ecological systems and shaped by teachers’ responses to the various predicaments that were typical of the ordinary course of the professional world. It was not only reflected in self-regulation at the psychological level and clear awareness of the life–work collision but also manifested the positive coping style at their behavioral level. Informed by the ecological systems theory [17,18], an ecological model for EFL teacher buoyancy in the online teaching setting was constructed, which unpacked the development of EFL teachers’ buoyancy in the process of their interaction with the ecological systems where they live (see Figure 1). Specifically, in the microsystem, faced with a series of problems brought about by online teaching (i.e., the lack of online teaching experience and ineffective classroom interaction, the unstable signal in the online class, poor immediacy of teaching, and the insufficient TPACK), teachers often resorted to psychological suggestion, autonomous learning, and online teaching training to enhance their professional and psychological development, and gained stronger buoyancy to tackle multiple teaching problems. Previous studies also indicated that striving for professional growth and coping with difficulties were the salient features of buoyant teachers [6,8]. Additionally, the activation of teachers’ adaptability and buoyancy might help them to navigate the new changes and uncertainties in their work and to display high-level engagement through positive self-adjustment [13]. In the mesosystem, teachers shouldered responsibilities posed by their multiple identities and easily suffered from long working hours and the blurred boundaries between home and school, particularly with the advent of the pandemic [5]. However, they endeavored to seek a balance between the work–life collision and found ways to address problems in the interaction with significant persons such as colleagues, students, and parents instead of avoiding those stressful realities [4]. In addition, the close relations between students and their parents in the mesosystem motivated them to show great concern about EFL teachers’ online teaching quality and efficiency and simultaneously expressed their understanding and support for the teaching work during the COVID-19 pandemic. In other words, some elements (i.e., relationships between EFL teachers and students’ parents, their colleagues, and school leaders) in the same mesosystem exerted a synergic and driving effect [4,5] on the development of teacher buoyancy. In the exosystem, changes in the local policies made by the educational authorities triggered the fluctuation of schools’ teaching requirements. Facing the challenges imposed by online teaching, EFL teachers with stronger buoyancy were able to exercise their active agency to adapt to the changing online teaching environment and seize the opportunities to successfully teach online and sustain students’ English learning motivation. Notably, as revealed from the interview excerpts in this study, the national policy related to online teaching in the macrosystem proposed higher demands for the local education department and school authority, who further made more specific requirements to push teachers to teach online and promote their buoyancy in the presence of teaching uncertainties. It also reflected that the dynamic interactions among different ecological systems and ecological factors within the same system [17,18] collectively contribute to the development of EFL teacher buoyancy.

## 6. Conclusions and Implications

The present study investigated the perceived challenges that EFL teachers encountered and the development of teacher buoyancy in their dynamic negotiation and interaction with multiple ecological factors in the technology-based educational environment. The findings in this study revealed that EFL teachers experienced multiple challenges in online teaching from different ecological systems. Specifically, these challenges included the lack of online teaching experience and ineffective classroom interaction, low teaching efficiency, and teachers’ inadequate TPACK in the microsystem, higher demands and expectations proposed by significant others in the mesosystem, heavy workload and work–life imbalance brought by requirements from schools and local education departments in the exosystem, the national educational policy and the COVID-19 pandemic in the macrosystem. They exerted significant impacts on the development of EFL teacher buoyancy; however, it is worth noting that there were also some protective factors in the ecological systems where they survived and thrived, such as support from colleagues, leaders, and students’ parents. That contextual support and spiritual encouragement contributed to teacher buoyancy [16] during the pandemic. EFL teachers, as developing and agentic agents, were not impotent against these adversities [1,3]. Their interconnection and interaction with the ecological systems were beneficial to shape and activate their buoyancy, enabling them to utilize resources to regulate their emotions and respond to stressful and challenging events in online teaching. Additionally, this study also made a constructive exploration of EFL teacher buoyancy from an ecological perspective and constructed the nested ecological systems model for EFL teacher buoyancy in online teaching settings. It revealed that EFL teachers’ buoyancy was shaped by the complex and dynamic interaction between teachers with ecological systems, and it was not only conceptualized as the individual’s ability but a socio-ecological product. It echoed the social-ecological view of resilience [27,45] which was usually regarded as one cognate term of buoyancy [7,21]. To a larger extent, this study expands our understanding of EFL teacher buoyancy from the psychological level to the ecological–psychological level and paves the way for future research on language teacher buoyancy or even language teacher psychology.

We found the buoyance of nine EFL high school teachers (short for teachers in this section) was formed in the interaction with different ecological systems; therefore, we propose the following implications from different levels. First, teachers, as developing persons with their unique life stories and capital [36], should have the awareness to regulate their emotions and appraise difficulties positively, endeavor to improve their technological pedagogical and content knowledge, and proactively cope with challenges, which are important features of buoyant teachers [6,16]. Teachers can empower themselves with new teaching skills and improve their technology integration competence [46] to overcome online teaching difficulties through school training programs in the mesosystem. Meanwhile, teachers can also actively seek multiple favorable interpersonal resources (i.e., the interaction and cooperation with significant others in the mesosystem) and familiarize themselves with the operation of the network technique. Third, teachers’ good relationships with significant others within and outside the school context [8] such as colleagues, school leaders, students, students’ parents, and family members, and their active engagement in these relational systems contribute to their online teaching buoyancy. In addition, schools should offer teachers more support and autonomy and establish an evaluation mechanism to facilitate teachers’ professional development and cooperation. More online and offline trainings of how to teach online are suggested to be organized by schools. These strategies may lead to positive feedback and engagement which is beneficial to the development of teacher buoyancy.

Although this study attempted to depict a complex picture of how EFL teachers’ buoyancy developed in their dynamic interaction with various ecological factors in the complex online educational context guided by Bronfenbrenner’s ecological systems theory, it has some limitations concerning the research participants and instruments that suggest directions for future research. First, only nine volunteers were selected as research participants in this study, and interview data was utilized to explore language teacher buoyancy in online teaching. It is hoped that future studies can enlarge the sample and employ multiple methods to collect and triangulate data. Researchers interested in this topic can also further demystify language teacher engagement in their interconnections with surrounding persons and events using a longitudinal study. Additionally, it may be a good attempt to use a mixed-method research design to develop and validate a scale to investigate the structure and level of language teacher buoyancy, and its relationships with other psycho-emotional traits (i.e., professional burnout, commitment, engagement, and job satisfaction) in an online teaching environment. To conclude, there is still much room for the exploration of language teacher buoyancy in the future.

## Figures and Tables

**Figure 1 ijerph-20-00613-f001:**
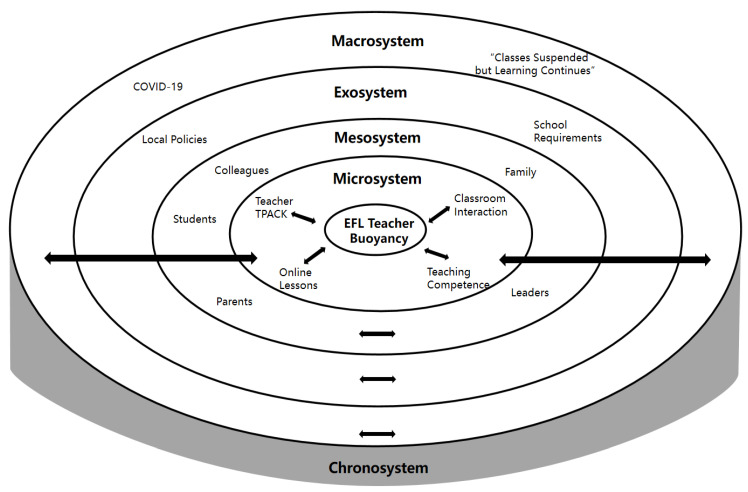
The nested ecological systems for EFL teacher buoyancy in the online teaching setting.

**Table 1 ijerph-20-00613-t001:** Basic information of the nine interviewees.

Participants	Gender	Age	Years of Teaching Experience	Educational Background	Social Media Platforms
T1	male	30	7	bachelor	Ding Talk
T2	female	31	7	bachelor	Tencent Meeting
T3	female	41	15	master	Tencent Meeting
T4	male	36	12	bachelor	Ding Talk
T5	female	30	4	master	Tencent Meeting
T6	female	35	13	bachelor	Tencent Meeting
T7	female	55	30	master	Ding Talk
T8	male	38	15	bachelor	Tencent Meeting
T9	male	31	6	master	Tencent Meeting

## Data Availability

The data presented in this study are available on request from the corresponding author. The data are not publicly available due to ethical considerations.

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
