# Peer review of "Unraveling EFL Teacher Buoyancy in Online Teaching: An Ecological Perspective"

_ijerph, 2022, doi:10.3390/ijerph20010613_

Round 1
Reviewer 1 Report
I believe this article is interesting and you have done an acceptable job with the background information and theory. However, there is an overuse of the word buoyancy, especially at the top of the second page. I realise you're listing types of buoyancy, but if there were ways to paraphrase parts of this section, it would be easier to read.
The sample is a convenience sample and is quite small. Since there are no inferential statistics, you cannot draw conclusions beyond this sample to the general population. Please make sure the conclusions are very specific to your sample and not to other teachers.
While linguists tend to look for results based on quantitative data and inferential statistics, it is more common in education studies to focus on qualitative results, which is why I believe this study would be of interest to the readers of this journal.
Author Response
Comment 1: I believe this article is interesting and you have done an acceptable job with the background information and theory. However, there is an overuse of the word buoyancy, especially at the top of the second page. I realise you're listing types of buoyancy, but if there were ways to paraphrase parts of this section, it would be easier to read.
Response 1: Thanks for this comment. We deleted some repetitive expression of “teacher buoyancy”. Meanwhile, we checked the whole paper and deleted these unnecessary and repetitive expressions.
Comment 2: The sample is a convenience sample and is quite small. Since there are no inferential statistics, you cannot draw conclusions beyond this sample to the general population. Please make sure the conclusions are very specific to your sample and not to other teachers.
Response 2: Thanks for this comment. We did some revisions in the implication part. In this part, we confine the implications to the “participants” of the study and prose some more practical implications. We deleted the last piece of suggestion “Fourth, local educational authorities should take into account the teachers’ teaching realities and setbacks happening in the immediate online teaching setting, organize necessary online teaching training seminars for teachers, and provide more resources to facilitate online teaching work.” We found it is a bit far from the participants.
While linguists tend to look for results based on quantitative data and inferential statistics, it is more common in education studies to focus on qualitative results, which is why I believe this study would be of interest to the readers of this journal.
Reviewer 2 Report
Some points in the discussion could be improved with reference to appropriate literature:
- See Benjamin Moorehouse's work on online teaching during covid
- See Sal Consoli's work on life capital for a stronger discussion of ecological perspectives
Author Response
Comment: Some points in the discussion could be improved with reference to appropriate literature:
- See Benjamin Moorehouse's work on online teaching during covid
- See Sal Consoli's work on life capital for a stronger discussion of ecological perspectives
Response: Thanks for this comment. The references you recommended were added in this paper. Here are these referencs.
Consoli, S. Life capital: An epistemic and methodological lens for TESOL research. TESOL Quarterly 2022, 56, 1397-1409.
Moorhouse, B. L., and Kohnke, L. (2021). Thriving or surviving emergency remote teaching necessitated by COVID-19: university teachers’ perspectives. Asia Pac. Educ. Res. 30, 279–287
Reviewer 3 Report
In the manuscript, there is reference to 'Tecent meeting', for example, in Table 4.1. Should this be 'Tencent meeting'?
Author Response
Comment: In the manuscript, there is reference to 'Tecent meeting', for example, in Table 4.1. Should this be 'Tencent meeting'?
Response: Thank you for this comment. All the wrong spellings of Tecent were corrected to “Tencent”.
Reviewer 4 Report
Review Comments for “Unraveling the EFL teacher buoyancy in online teaching: An eco-logical perspective”
Overall, this manuscript is clearly written and can be considered to be published after addressing the review comments below. At the same time, a professional language editing is necessary because there are language issues.
Abstract: Please add the year for Bronfenbrenner
Introduction: 1) What is the difference between teacher buoyance and teacher agency. Both seem same based on the authors’ definition. 2) Relevant reference support is needed for “teacher buoyancy to date in the online teaching setting is still at an under-explored stage”. 3) Why did the research focus on language teachers? Related research gap needs to be addressed.
Literature review: 1) Please try to cut down using “some” throughout the manuscript, which is not clear. 2) Why mentioned teacher education field? Does this research relate to preservice teachers? 3) Is online teaching an acute or everyday challenge for teachers? It is hard to define. 4) Why did the authors focus on EFL teachers? The logic connection among paragraphs is not clear and needs to be addressed. 5) Relevant year is needed for Bronfenbrenner in 2.2.
Bronfenbrenner’s ecological systems theory: 1) Why did the ecological systems theory relate to online teaching? This is not well addressed yet. 2) A figure here can help readers to understand this theory.
Methodology: 1) Please revise “settings” to be “research context”. 2) Why participants’ differences need to be considered? 3) What were the criteria for the participant selection? 4) Please indicate the research ethical approval, which is a fundamental issue. 5) Why were Semi-structured interviews used in this research? 6) Why was the reflection needed in the interviewing? 6) What is “technological pedagogical and content knowledge (TPACK)”? The jargon needs to be addressed. 7) Details of data analysis are needed, such as coding examples.
Findings: 1) Most cited accounts from the participants seem to long, and please cut down irrelevant wordings. 2) The current finding needs to be enhanced for its weakness. In particular, there were few references cited in this part to connect and interpret the finding. In this regard, the finding can become subjective and lack trustworthiness.
Please add the discussion part to be complete.
Conclusion and implications: 1) Please make the implications to be more specific and practical. The current ones are still much general. 2) If the authors would develop and validate a scale, why was the research gap of qualitative method mentioned? A mix-method perspective can be better, not only the scale.
Author Response
Thank you for the suggestions and comments from Reviewer 4. We put all our responses into a word file and uploaded it here.

Reviewer 5 Report
This qualitative study concentrated one major question, that is, how EFL teachers shaped and exercised buoyancy in their negotiation with different ecological systems in online teaching guided from an ecological perspective. It was further developed into two sub-questions, namely, the perceived challenges they have encountered and the development of teacher buoyancy in their dynamic negotiation and interaction with ecological systems. All these questions aimed to help the readers to make clear how the sampled Chinese EFL teachers coped with challenges and adversities, and became buoyant in online teaching.
The topic is original and much too related in the field of language teacher education. Buoyancy has been focused in the field of education in the past ten years, however, it was less researched in language teacher education. Language teachers face many challenges from the macro social and cultural contexts, and micro schools, institutions, colleagues, and communities. Some scholars even warn that being language teachers is a highly demanding career. In fact, the current study addressed a salient factor related to this demand, namely, language teacher buoyancy (LTB). To my best knowledge in the field of language teacher psychology, the limited number of studies on teacher buoyancy were quantitative. Less attention was paid to the dynamic change and development of LTB using a qualitative research design. This study offers an insight into how EFL teachers deal with the online teaching setbacks and the research on language teacher psychology from the ecological perspective.
Compared with other published materials, as I said in Point 1 and 2, the current study, primarily delved into the development of EFL teachers’buoyancy in their dynamic interaction with ecological systems where they lived, which was rarely done. Specifically, ecological perspective gives an inspiration on how their buoyancy affects or is affected by environments.
In regard to methodology, I do think this research has followed a strict qualitative way to unfold the research participants, instruments, and procedures. I didn’t find much to be improved. I think the conclusions are consistent with the evidence and arguments. The research questions have been well addressed. The current references are all closely related to the topic. However, I suggested the authors can update the new publications on teacher buoyancy. They may go to some top-ranking journals to look for some new references.
I hope that the authors can add some new publications in the past two years in the literature review part. Also, the authors should follow the format of manuscript of journal to revise the paper.
Author Response
I hope that the authors can add some new publications in the past two years in the literature review part. Also, the authors should follow the format of manuscript of journal to revise the paper.
Response: Thank you for the comments. We have updated the literature and added more related references. Here are some pieces of literature.
- Tang, S.Y.; Wong, A.K.; Li, D.D.; Cheng, M.M. Teacher buoyancy: Harnessing personal and contextual resources in the face of everyday challenges in early career teachers’ work. Eur. J. Teach. Educ. 2022, doi:10.1080/02619768.2022.2097064.
- Gong, Y.; Fan, C.W.; Wang, C. Teacher agency in adapting to online teaching during COVID-19: A case study on teachers of Chinese as an additional language in Macau. Journal of Technology and Chinese Language Teaching 2021, 12, 82-101.
- McGarr, O.; Passy, R.; Murray, J.; Liu, H. Continuity, change and challenge: Unearthing the (fr)agility of teacher education. Journal of Education for Teaching 2022, 48, 490-504.
- Huang, S. A review of the relationship between EFL teachers’ academic buoyancy, ambiguity tolerance, and hopelessness. Front. Psychol 2022, 13, 831258.
- Li, M. On the role of psychological health and buoyancy in EFL teachers' professional commitment. Front. Psychol. 2022, 13, 897488.
- Toom, A.; Pyhältö, K.; Rust, F.O.C. Teachers’ professional agency in contradictory times. Teach. Teach. 2015, 21, 615-623.
- Kayi-Aydar, H. Language teacher agency: Major theoretical considerations, conceptualizations and methodological choices. In Theorizing and Analyzing Language Teacher Agency, Kayi-Aydar, H., Gao, X., Miller, E.R., Varghese, M., Vitanova, G., Eds.; Multilingual Matters: Bristol, UK, 2019; pp. 10-23.
- Tao, J.; Gao, X.S. Language Teacher Agency; Cambridge University Press: Cambridge, UK, 2021.
- Ding, J.; He, L. On the association between Chinese EFL teachers’ academic buoyancy, self-efficacy, and burnout. Front. Psychol. 2022, 13, 947434.
- Elahi Shirvan, M.; Rahmani, S.; Sorayyaee, L. On the exploration of the ecology of English language teachers’ personal styles in Iran. Asian. J. Second. Foreign. Lang. Educ. 2016, 1, 12.
- Consoli, S. Life capital: An epistemic and methodological lens for TESOL research. TESOL Quarterly 2022, 56, 1397-1409.
- Liu, Y.; Zhao, L.; Su, Y.S. The impact of teacher competence in online teaching on perceived online learning outcomes during the COVID-19 outbreak: A moderated-mediation model of teacher resilience and age. Int. J. Environ. Res. Public Health 2022, 19, 6282.
- Liu, H.; McGarr, O.; Murray, J.; Passy, R. Learning from Covid-19-continuity or change in teacher education? Journal of Education for Teaching 2022, 48, 389-392.
- Gu, Q.; Li, Q. Sustaining resilience in times of change: Stories from Chinese teachers. Asia-Pac. J. Teach. Edu. 2013, 41, 288-303.
- Gong, Y.F.; Lai, C.; Gao, X.S. Language teachers’ identity in teaching intercultural communicative competence. Language, Culture and Curriculum 2021, 35, 134-150.
- Gu, Q. (Re)conceptualising teacher resilience: A social-ecological approach to understanding teachers’ professional worlds. In Resilience in Education Concepts, Contexts and Connections, Wosnitza, M., Peixoto, F., Beltman, S., Mansfield, C.F., Eds.; Springer: New York, NY, USA, 2018; pp. 13–33.
- Gong, Y.; Gao, X.; Lai, C. Novice teachers’ technology integration and professional identity reframing in the Chinese as an additional language classroom. In Theory and Practice in Second Language Teacher Identity: Researching, Theorising and Enacting, Sadeghi, K., Ghaderi, F., Eds.; Springer: New York, NY, USA, 2022; Volume 57, pp. 195-210.